# Teaching in Secondary Education Teacher Training with a Hybrid Model: Students' Perceptions

José Luis Martín-Núñez *[iD], Juan Luis Bravo-Ramos [iD], Susana Sastre-Merino [iD], Iciar Pablo-Lerchundi, Arturo Caravantes Redondo and Cristina Núñez-del-Río

Institute of Educational Sciences, Technical University of Madrid, 28040 Madrid, Spain;
juanluis.bravo@upm.es (J.L.B.-R.); susana.sastre@upm.es (S.S.-M.); iciar.depablo@upm.es (I.P.-L.);
arturo.caravantes@upm.es (A.C.R.); mc.nunez@upm.es (C.N.-d.-R.)
* Correspondence: joseluis.martinn@upm.es

**Abstract:** The exceptional situation experienced on the planet during the last two academic years has forced teachers to adapt their teaching. They had to pass their face-to-face teaching methodologies to online teaching. This study analyzed the perceptions of students in a master's degree in secondary school teacher training, who have experienced the transition to bimodal teaching during the 2020–21 and 2021–22 academic years carried out by professors with high digital teaching competence. Data from two academic years are available, with a total of 207 students. The results indicate that the experience has been positive, and no significant differences were found in the follow-up of the contents when the class was followed online. However, the feeling of belonging to the group is lower in online classes than in face-to-face classes, and the perception of online teaching is more favorable in the academic year immediately developed after the COVID crisis. The bimodal modality has become a useful solution as long as the conditions of the equipment and networks are guaranteed as well as the training of teachers in digital teaching competence. In addition, the students' perception of this modality is that it can continue once the exceptionality of the pandemic has been overcome.

**Keywords:** digital teaching competence; online-training; hybrid teaching; students' perceptions

## 1. Introduction

During the second semester of the 2019–20 academic year, the world changed dramatically due to the COVID-19 pandemic. In a very short time, drastic decisions were taken, as the medical authorities lacked the necessary speed of action, and the health services were not prepared to afront such a global crisis. In the education system, the result was the closing of schools, high schools, and universities. Teachers were advised to continue teaching from their homes, without guidelines, previous training, and above all, resources [1]. Therefore, because of this extreme situation, restrictions and confinements were suddenly part of everyday life and had very clear consequences on education. Traditional face-to-face classes, which were, at the most, partly supported by a virtual classroom, had to go completely online overnight [2].

Different studies have described the difficulties experienced both by teachers and students, such as the lack of a "home office" infrastructure, like computers, internet availability, and enough available bandwidth. The differences in access have broadened the divide to receive quality education, which was already alarming and put into world development agencies' agendas [3]. In addition, teachers need time and effort to adapt the classes, and they experience a lack of information and communications technology (ICT) and pedagogical skills to professionally design and deliver virtual education online [4–6]. Schools and other academic centers in all education levels did not have the means and resources to provide their teachers with the necessary tools to carry out their jobs [7]. García García stated that the education systems of different countries established online teaching,

forcing their teachers to change their teaching method overnight from face-to-face to distance teaching [8]. Synchronous and asynchronous lessons were organized in a completely new setting: the homes, both of students and teachers. In this context, challenges related to multiple distractions and connectivity were present. Therefore, tools and abilities to counteract them became essential [8].

Teachers used to face-to-face teaching had to learn overnight the typical aspects of online teaching, such as the use of ICT, especially videoconference software, all at a high personal cost. As [9] pointed out, the teaching that could be offered in this first moment can be considered emergency remote teaching (ERT). This is not a carefully planned teaching to be offered online with all quality standards. From this time, we can find multiple opinions. Some teachers perceived that they could solve the situation quiet well, while others had a negative experience, reflected in a defeatist opinion about the use of online teaching methodologies. The lack of interpersonal contact and the change of the learning scenario (from the classroom to the homes) comprised the main difficulties in a confinement situation [10]. Personal interaction was replaced by a screen, which is an arduous and sometimes difficult task. In an online teaching setting, teachers need to assume a great effort to check students' attention and understanding, as students do not frequently turn on their camera. The lack of visual contact gives teachers a sensation of talking alone [7,11].

Nevertheless, good teaching practices to improve online teaching have also been found. These are applicable to distance teaching and also to face-to-face teaching, where video lessons have become an essential support in the hybrid model. Many teachers see this exceptional situation as an opportunity to explore more flexible methodologies, such as mixed or hybrid learning or the alternance between synchronous and asynchronous learning [12].

The new situation in the education field is not unique. In all professional sectors, home office and digitalization burst into everyday work, generating a completely new model of work and personal interactions. Public opinion seems to become aware that the increase of digitalization could be a good preventive strategy for any unexpected crisis in general and, specifically, in the education sector. Therefore, this model will be established from now on and into the future [13]. The pandemic has clearly accelerated the digitalization process in the education sector. At present, a great part of teaching takes place in an online or partially online setting. In higher education, the methodological and technical change was already a challenge for the 21st century society, but the pandemic has become a real trigger for it [14].

## 2. Literature Review

### 2.1. Digital Teaching Competence

Since 2010, the introduction of the European Higher Education Area (EHEA) has prompted the use of ICTs in the university classroom. Therefore, a part of the faculty did not start from scratch concerning their digital teaching competence (DTC) when they had to change to online teaching. Nevertheless, the pandemic revealed the digital divide of many professors who had digital tools at their disposal but did not use them or used tools such as learning management systems (LMS) only as a repository for class materials [7]. In fact, following [2], although 95.74% of Spanish universities have an LMS, the quality of online teaching depends on the pedagogical design because technology on its own is no guarantee of success. Professors need to master virtual environments, including material management, tasks and activities creation, and interaction with students, through all available communication channels and have an active role as guides in the teaching–learning process by combining different digital tools [2].

In order to achieve success in online activities, certain conditions must be fulfilled according to [15]: (a) responding to participants' extrinsic motivations; (b) including content that causes intellectual stimulus, engagement, and, therefore, learning results; and (c) designing tasks and teaching strategies that humanize the assessment, trying to keep technological mediation to a minimum and fostering interpersonal relations.

Following [16], it is important to consider a series of recommendations to maintain the effectiveness of online classes. For this, she established aspects to check before, during, and after each online learning activity as follows:

1.  Before the activity, it is relevant to share the agenda with the students to communicate the expectations for the class. This way, they know what they are presumed to learn, and they can prepare for the class beforehand. In addition, the interest of students to connect to the class must be addressed, and the best way to do it is by presenting the content as interesting and attractive. In this sense, visual resources, short explanations, graphs, etc., are a useful tool to trigger students' attention, so these must be prepared by the teacher. If students can visualize the content, they will be more focused and follow the class without difficulties.

2.  During class, it is highly recommended to connect some minutes before schedule to chat with students as they start to enter the virtual classroom. This is the same thing a teacher does in the actual classroom while students start to come in, and it creates a sense of sympathy that inclines students to pay attention to the theoretical explanations. This would also be the moment to ask them to turn on their cameras as long as the technical conditions permit it and in order to facilitate communication and active learning similar to an actual classroom environment. This would minimize the sense of inability to receive feedback from teacher and students due to lack of eye or personal contact, enabling the teacher to develop rapport and establish friendly, supportive relationships [7,17]. The first part of class should be devoted to recall the agenda so that students get a clear idea of the class content and activities. Frequent questions during class help to maintain an active interest from students either through direct dialogue or through the chat, by using emoticons, online polls, or just-in-time questionnaires [18]. Moreover, one of the most effective ways to enhance student interaction is using virtual whiteboards, where many people can interact, write, share hyperlinks, images, or videos, and the results can be saved for later review [18]. The authors also propose the use of "breakout rooms" for small collaborative learning. Finally, time control is crucial. Therefore, teachers must be concise, maintain an active pace, and keep the classes as short as possible. They must be aware that a student is able to keep attention longer in a face-to-face class than in an online class. To sum up, as [18] stated, teaching and cognitive and social presence must be considered for an effective synchronous online educational experience.

3.  After the class, teachers must collect the students' feedback through different tools, such as surveys, to measure the effectiveness and receive suggestions. As [19] stated, we would be committing a great error if we do not learn from our experience and our mistakes. We need to confront the next class with a proper reflection and planning. In sum, the end of a class should allow us to receive information about its development and the students' understanding and learning to confront the next class with a proper preparation.

Although ICTs are nowadays more present among the teaching competences [20], they are only acquired through voluntary training or autodidactically. The COVID-19 pandemic brings to light the necessity to achieve these competences in all teachers. In order to provide students with the necessary resources for their learning, achieving these competences for teachers has now turned into an obligation [21]. Undoubtedly, the digital divide has been reduced among teachers with the crisis, as most of them are now aware of the advantages (but also the difficulties) of using ICTs in education. The digital transformations offer innumerable options, which foster rethinking the teaching profession [7].

Many frameworks have been created around the "digital competence" concept both for students and teachers. They are centered on informational and didactic competences. One of the best-known frameworks is the "European Framework for the digital competence of educators. DigCompEdu" [22]. It establishes six levels in digital competence [14]:

1.  Newcomers: teachers who are not familiar with the use of digital tools;
2.  Explorers: teachers who have just begun using digital tools;

3. Integrators: who use digital tools in diverse contexts and for varied goals;
4. Experts: who use digital tools with confidence, creativity, and critical thinking to improve their performance;
5. Leaders: teachers who use a broad repertoire of flexible, complete, and effective digital strategies;
6. Pioneers: they question the suitability of digital and pedagogical practices in which they are proficient. At the same time, they are leading innovators and models for other teachers.

The digital competence of the university teacher has been evaluated [23,24] and contemplated in broader models as one of the essential aspects in teaching skills [25]. As part of it, the effective use of digital technologies by teachers implies relating basic digital skills with the didactic use of technology and the ability for continuous learning, making students aware of this [26,27]. However, digital teaching competence must go beyond this, providing continuous diagnostic assessment and including educational resources for its development [28].

*2.2. Teaching Modalities*

The new educational services offered by learning platforms, and especially the arrival of COVID-19, led universities worldwide to implement changes in their teaching modalities. Specifically, the Universidad Politécnica de Madrid (UPM) promoted two ICTs-based education modalities: bimodal synchronous education, which combines face-to-face and distance learning, and blended learning, in which asynchronous activities are carried out through LMSs. These models try to efficiently combine face-to-face and distance activities taking into account the advantages according to the content and learning objectives.

The urgency of the COVID lockdown led to an emergency remote teaching (ERT) in which face-to-face classes where simply moved to an equivalent remote synchronous class through video conferencing systems. When educational centers could be opened again, the synchronous education model was maintained but in a hybrid modality [29], in which part of the students attend the class in the actual classroom, while another part follows it through a videoconferencing system. They can attend class individually from home or jointly from a mirror classroom, that is, a classroom which is close to the actual classroom where the professor teaches and where a group of students may follow the session through videoconference.

Throughout the article, we use the term "hybrid" for this type of synchronous bimodal (face-to-face and remote) sessions. The issue is to combine the design of face-to-face and online sessions in a single model that integrates all students in the same learning space.

Taking into account the learning efficiency, the main goal of learning technologies is to promote flexibility, such as these hybrid educational models offer [30]. Students choose their way of approaching learning concerning different aspects, such as timing (videoconference/materials), interaction (face to face/remote), and location (physical/virtual).

*2.3. Students' Perceptions*

Different studies have analyzed students' perceptions during the pandemic. Some affirm that students perceive transition to e-learning as a useful solution in view of the medical crisis [31]. However, other findings indicate that the quality of technical service should be improved, specifically teacher training, and teachers should be encouraged to use strategies that allow student participation [32]. An adequate infrastructure and the quality of the interactions influence students' perception of their learning process in online teaching [33,34].

In the case of Science, Technology, Engineering, and Mathematics (STEM) students, a study in a Spanish university found that a majority was not satisfied with the quality of their online education and that it negatively affected their academic performance [35]. STEM students at Lund University (Sweden) were also less satisfied with their courses compared to previous face-to-face cohorts due to receiving less feedback and fewer valuable

comments. They found it harder to understand the expectations and standards of work [36]. Another study at California State University (USA) identified various issues that negatively influenced online engineering education, including logistical/technical problems, learning/teaching challenges, privacy and security concerns, and lack of sufficient hands-on training [37].

Benefits of online teaching for students have been thoroughly studied for years. One of the most salient ones is that it offers a flexible training that facilitates the interaction between students and teacher and among students [38,39]. If videos are used, this effect is enhanced, as they create a learning environment propitious for good social interaction [40]. Nevertheless, the pandemic is a different scenario that clearly influences students' psyche. Their behavior towards online teaching is conditioned by stress, and many unknown factors are at work [41]. Some studies have shown that one of the stressors was their uncertainty over the end of semester exams and assessments [42] or the required load of work [43]. Another study reported students feeling discouragement, boredom, confusion, and worry to a greater extent and calm and trust to a lesser extent, all of them related to the greater or lesser connection with classmates and teachers [35]. Hence, students' perceptions are greatly connected to the pedagogical competences of their teachers as well as the class design and the connections and communication mechanism among students and between teachers and students [35].

### 2.4. Research Questions and Objectives

This work considers students' learning during different teaching adaptations since the pandemic started. Teaching modalities have changed from completely face-to-face to completely online and then again to a hybrid model. The article analyzes the results of adjusting teaching to the "new normal" scenario, in which half of the students in one class receive their lesson face-to-face, while the other half stays at home and follows it online. Each week they change places so that both sub-classes experience both teaching modalities. The students that participated in these classes are future secondary education teachers.

Taking this context into account, the research questions of this study are: (1) How do students perceive their learning experience when comparing face-to-face and online synchronous classes? (2) Are there differences in students' perceptions of the hybrid model from the year 2020–21 (first year with the hybrid model) to 2021–22 (second year)?

### 3. Materials and Methods

During the second semester of the 2019–20 academic year, when the COVID-19 pandemic broke out, educators tried to accomplish the challenge of keeping a fluent communication with their students and finishing the academic year in the best possible way.

We present an experience that started during the next academic year (2020–21). At that moment, fear and the interference of the necessary measures taken by the university were at the epicenter of our work. Teaching was addressed in better conditions than before but with an enormous uncertainty. Our university supported a hybrid teaching model, by which part of the students attended class at the university, and the other part followed the same class through an online platform. We organized it by dividing each class into two sub-classes so that every group had face-to-face class every 15 days. Hence, each student attended face-to-face class one week and had online class the next week.

The evolution of the pandemic and the improvement of the medical situation allowed a better development for the 2021–22 academic year. On one hand, face-to-face classes increased. On the other hand, technological means offered the possibility to provide online classes for those students who needed it (confinements, illness, or other justified reasons that prevent attending class).

To guarantee the success of an online class, the DTC of the faculty must be well developed, but they also need the appropriate technical equipment (see Figure 1). These technical tools, directed towards the recording of sound and image and its broadcast, allow for a fluent communication between participants in the classroom and at home.

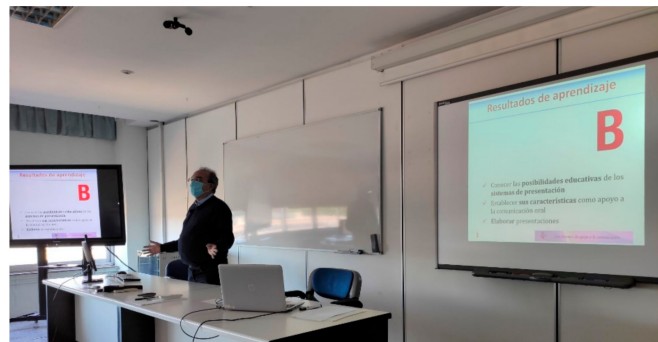

**Figure 1.** Classroom equipped with audiovisual resources that allow a high-quality broadcast.

Our study refers to the Master's Degree in Secondary Education (MSE) of the Universidad Politécnica de Madrid (Technical University of Madrid). It is structured in three modules: General, Specific and Practicum. The analyzed experience takes places in the subject entitled "Resources and Technical Support to Communications"—RTSC. It is a compulsory subject in the general module that takes ten weeks of class and entails three ECTS credits.

### 3.1. Participants

The study's sample is comprised of the students enrolled in the RTSC subject in the four specializations that conform the Scientific-Technological itinerary of the MSE. These are: Technology (TC), Mathematics (MAT), Physics and Chemistry (P&C), and Graphic Expression (GE). In the 2020–21 academic year, 95 students out of 111 active students filled out a questionnaire at the end of the 10-week class period of RTSC. This corresponds to 86.3% of responses. In the 2021–22 academic year, 77 students out of 96 (80.2%) responded to the same survey. In both cases, the participation level may be considered very high. Table 1 shows the students' distribution by specialization and gender.

Two professors with a high DTC, who could be labeled as Leaders or Pioneers [14], taught RTSC in both academic years. They worked in a collaborative manner, carefully planning their classes and making the required adaptations for the singular conditions in which teaching had to take place.

**Table 1.** Students' distribution by specialization and gender.

|  | TC | | | MAT | | | P&C | | | GE | | | Total | | |
| --- | --- | --- | --- | --- | --- | --- | --- | --- | --- | --- | --- | --- | --- | --- | --- |
|  | M | F | T | M | F | T | M | F | T | M | F | T | M | F | T |
| 2020–21 | 14 | 11 | 25 | 15 | 12 | 27 | 11 | 18 | 29 | 13 | 17 | 30 | 53 | 58 | 111 |
| 2021–22 | 14 | 8 | 22 | 8 | 14 | 22 | 16 | 9 | 25 | 13 | 14 | 27 | 51 | 45 | 96 |

TC, Technology; MAT, Mathematics; P&C, Physics and Chemistry; GE, Graphic Expression; M, male; F, female; T, total.

### 3.2. Instruments

The students answered a short survey about the teaching methodology used for the specific setting in which the subject had to be taught during the years 2020–21 and 2021–22. This survey consists of eight items for the 2020–21 year and six in 2021–22, as two items were removed due to the return to face-to-face class. All present a six-point Likert scale from completely agree (6) to completely disagree (1). Acceptable reliability coefficients were obtained (r = 0.780 and r = 0.669, respectively).

These items, designed ad hoc, following the guidelines of [44], refer to different aspects about online teaching compared to face-to-face teaching, namely:

1. Following the sessions through online classes without difficulties;
2. Following online classes the same way as face-to-face classes;
3. Doing proposed activities during class;

4.    Creating a sense of belonging to the group;
5.    Participating in class;
6.    Maintaining the subject's pace;
7.    Assessing the hybrid model as a solution beyond COVID-19;
8.    Having a transmission without technical difficulties.

### 3.3. Procedure

Some authors [28,31,33,38,40,41,45,46] indicated that the pandemic has had a clear influence on university teaching. In relation to DTC, these authors also agreed that the pandemic has created the necessity of including digital tools to continue online classes. This was our starting point.

Our teaching team believes that a master's degree, such as ours, which prepares its students to be teachers who will essentially teach face-to-face classes, should not be taught completely through online classes. Nevertheless, classes could not be entirely face-to-face, as anti-COVID regulations compelled to reduce the number of students present in class. To achieve good-quality classes in this unfavorable situation, we had to take advantage of the digital teaching competences that were fully developed in most of the professors that teach the general module of the master's program. Taking this into account, we were convinced that, such as [19] affirmed, methodological decisions should prevail over technological decisions and not the other way around.

The teaching–learning methodology common to the whole general module focuses on active learning. Students work in all subjects in heterogenous groups formed by classmates of the four specializations. Whenever possible, these groups are composed by four students, each belonging to one of the specializations. Gender is also taken into account, attempting to ensure males and females are equally represented in each group. With this organization, students are distributed in three classes (A, B, and C) of approximately 40 students. Therefore, each class is composed of ten groups.

During the 2020–21 academic year, class attendance was limited to 20 students. In this context, a hybrid teaching model was established. Each week, five of the ten groups that composed a class attended face-to-face class, while the rest attended the same class online. The next week, they exchanged places so that the students who had stayed home, now attended face-to-face classes, while the rest received the same class online. This way, each student attended five face-to-face classes and five online classes for each subject, as all subjects were organized in a ten-week period.

In the 2021–22 academic year, due to the improvement of the medical situation, conditions changed. Now, face-to-face classes could be established for all students in the third week of class. Therefore, students attended two weeks of class with the hybrid model (one week face-to-face and one week online). Afterwards, only justified cases had the chance to attend the classes online, while their classmates were at the university. This means that in the 2021–22 academic year, students answered the survey with their participation in the hybrid model in 20% of the classes, while in the previous year, they had experienced this model in a 100% of the classes. For this reason, items 2 and 3 were eliminated from the study for year 2021–22.

The RTSC subject is related to the use of ICTs for teaching. As both professors had received training in DTC, they could follow a pedagogical methodology similar to that proposed by [16]. In both years, each class followed the structure depicted in Table 2.

**Table 2.** RTSC class structure during academic years 2020–21 and 2021–22.

| Phase | Description |
|---|---|
| 1. Connection | Ten minutes before the beginning of the class, the professors connected with the groups that had their class scheduled for online teaching. This connection was established through BB-Collaborate in the 2020–21 academic year and with Zoom in 2021–22. In both cases, it was easy to connect for students, as they could access the videoconference room through the LMS (Moodle). The videoconference room for the class was one of the technical resources, and it permitted the integration of other technical tools that could be used during class. |
| 2. Verification | The proper functioning of the system had to be checked. It is based on:<br><br>(a) Connection software (BB-Collaborate and Zoom);<br>(b) Fixed camera situated at the back of the classroom with remote control to establish different positions depending on where the professor stands or what resources he uses: digital interactive screen, interaction with students, whiteboard, etc.;<br>(c) Personal microphone attached to the professors' clothes, environment microphones that allow the students at home to listen to the classes' sounds, and hand-microphone (properly disinfected after each use) that allows students who assist online to listen properly to their classmates' interventions. |
| 3. Group belonging | At the start of the lesson, interaction between the students in the classroom and those online is encouraged so that all feel they are part of the same class. |
| 4. Lesson | The lesson is taught following different methodologies:<br><br>(a) Videos that students must see prior to class (adapted flipped classroom);<br>(b) Theoretical explanation based on different resources, such as presentations (different formats, e.g., PowerPoint, Canva, Genially, Prezi, etc.) and other pedagogical tools;<br>(c) Reading of SCORM documents available in Moodle. |
| 5. Practical tasks | Students participate in the class in different active tasks that are then presented to all their classmates (present in the classroom or online):<br><br>(a) Use of various presentation software (Canva, Genially, Prezi, etc.) to individually create content;<br>(b) Practice with video-recording software and creating a video in groups to be exhibited;<br>(c) Ten-minute mini lessons taught by the students, who receive feedback from their classmates through commentaries and also through Mentimeter questionnaires. |
| 6. Questionnaires | Brief questionnaires using Kahoot, which are part of the subject's assessment. They are carried out at the end of each theoretical content and at the start of those lessons that required students to watch a video prior to class. |
| 7. Communication | Faculty encourage communication with all students: those present in the classroom but, especially, those who attend the class online. For this, professors ask questions, make proposals, instigate reflection, etc. Students at home may answer directly using their microphone, raising their hand in the virtual classroom, or writing in the chat. For this last option, and in order to be aware of what students actually see, the professor in class has a second screen where he could see the same as the online students and read the chat. |

### 3.4. Design

This study uses a quantitative design of pre-experimental nature, with no control group. The groups usually used in the general module of the master's program were an optimal choice to organize the hybrid learning model introduced in the 2020–21 academic

year and were again used in 2021–22. Data were collected each year at the end of the RTSC subject.

### 3.5. Data Analyses

The statistical analyses were carried out using the SPSS for Windows, version 26. To assess differences, ANOVA's F-tests were carried out for independent samples.

## 4. Results

This section is organized in the following way: First, data obtained in each academic year are presented. Then, data are compared to assess if there are significant differences in the students' perception depending on the year. Sections 4.1 and 4.2 show the survey's results disaggregated by academic year, specialization, and class group.

### 4.1. Results for 2020–21 Academic Year

A descriptive reading of the data obtained in the 2020–21 academic year reveals that students had, in general, a positive and high opinion on the six-point Likert scale about the online classes. They did not relate difficulties concerning the class' pace nor serious technical problems. Considering item per item, we established the following assertions (Table 3):

**Table 3.** Results of the survey by specialization. Mean (standard deviation) and Snedecor's F (ANOVA).

| Item | Year | TC | MAT | P&C | GE | F<br>p | Total | F<br>p |
|---|---|---|---|---|---|---|---|---|
| Item 1. Online classes allowed me to follow the session without difficulties | 20–21 | 5.15 (0.99)<br>N = 20 | 5.46 (0.88)<br>N = 24 | 5.44 (0.89)<br>N = 27 | 5.38 (0.92)<br>N = 24 | 0.516<br>0.673 | 5.37 (0.91)<br>N = 95 | 28.88<br>0.000 *** |
| | 21–22 | 4.29 (1.52)<br>N = 21 | 4.53 (1.12)<br>N = 17 | 4.58 (1.35)<br>N = 19 | 4.40 (1.39)<br>N = 20 | 0.185<br>0.906 | 4.44 (1.34)<br>N = 77 | |
| Item 2. Online classes allowed me to follow the class the same way as face-to-face classes. | 20–21 | 4.35 (1.57)<br>N = 20 | 4.83 (1.37)<br>N = 24 | 4.89 (1.4)<br>N = 27 | 5.00 (1.06)<br>N = 24 | 0.954<br>0.418 | 4.79 (1.35)<br>N = 95 | Not applied in 21-22 |
| | 21–22 | | | | Not applied | | | |
| Item 3. In online classes, I was able to do all the proposed activities during class. | 20–21 | 4.60 (1.31)<br>N = 20 | 5.04 (1.12)<br>N = 24 | 5.33 (0.92)<br>N = 27 | 5.04 (0.62)<br>N = 24 | 2.025<br>0.116 | 5.03 (1.03)<br>N = 95 | Not applied in 21-22 |
| | 21–22 | | | | Not applied | | | |
| Item 4. In online classes, I was able to feel a sense of belonging to the group. | 20–21 | 4.20 (1.36)<br>N = 20 | 5.38 (0.92)<br>N = 24 | 5.26 (1.02)<br>N = 27 | 5.17 (0.96)<br>N = 24 | 5.427<br>0.002 ** | 5.04 (1.14)<br>N = 95 | 23.06<br>0.000 *** |
| | 21–22 | 3.57 (1.47)<br>N = 21 | 4.24 (1.30)<br>N = 17 | 4.26 (1.33)<br>N = 19 | 4.53 (1.12)<br>N = 19 | 1.938<br>0.131 | 4.13 (1.34)<br>N = 75 | |
| Item 5. In online classes, I felt more confident to participate in class. | 20–21 | 4.00 (1.45)<br>N = 20 | 4.92 (1.1)<br>N = 24 | 4.52 (1.37)<br>N = 27 | 4.92 (0.97)<br>N = 24 | 2.670<br>0.052 | 4.61 (1.27)<br>N = 95 | 24.36<br>0.000 *** |
| | 21–22 | 3.10 (1.25)<br>N = 20 | 3.06 (1.56)<br>N = 17 | 3.89 (1.52)<br>N = 19 | 4.21 (1.32)<br>N = 19 | 3.117<br>0.031 * | 3.57 (1.47)<br>N = 76 | |
| Item 6. Following face-to-face classes with the participation of classmates from home hinders the class' pace. | 20–21 | 2.15 (1.39)<br>N = 20 | 2.96 (1.94)<br>N = 24 | 3.19 (1.69)<br>N = 27 | 2.83 (1.61)<br>N = 24 | 1.540<br>0.210 | 2.82 (1.69)<br>N = 95 | 13.57<br>0.000 *** |
| | 21–22 | 3.71 (1.38)<br>N = 21 | 4.41 (1.58)<br>N = 17 | 3.11 (1.56)<br>N = 19 | 3.79 (1.36)<br>N = 19 | 2.377<br>0.077 | 3.74 (1.51)<br>N = 76 | |
| Item 7. Online classes are only a solution for the specific situation we are living in now. | 20–21 | 3.40 (2.09)<br>N = 20 | 3.96 (2.05)<br>N = 24 | 4.52 (1.58)<br>N = 27 | 3.33 (1.79)<br>N = 24 | 2.176<br>0.096 | 3.84 (1.9)<br>N = 95 | 3.73<br>0.055 |
| | 21–22 | 4.43 (1.69)<br>N = 21 | 4.24 (1.82)<br>N = 17 | 4.32 (1.45)<br>N = 19 | 4.47 (1.50)<br>N = 19 | 0.081<br>0.970 | 4.37 (1.59)<br>N = 76 | |
| Item 8. During online classes, I had technical or other problems that hindered me following the class. | 20–21 | 2.50 (1.57)<br>N = 20 | 2.42 (1.79)<br>N = 24 | 2.33 (1.75)<br>N = 27 | 2.54 (1.84)<br>N = 24 | 0.070<br>0.976 | 2.44 (1.72)<br>N = 95 | 0.94<br>0.334 |
| | 21–22 | 2.19 (1.44)<br>N = 21 | 3.12 (1.87)<br>N = 17 | 2.63 (1.57)<br>N = 19 | 2.95 (1.90)<br>N = 19 | 1.124<br>0.345 | 2.70 (1.70)<br>N = 76 | |

TC, Technology; MAT, Mathematics; P&C, Physics and Chemistry; GE, Graphic Expression. * $p \leq 0.05$; ** $p \leq 0.01$; *** $p \leq 0.001$.

The online classes allowed students to follow the lessons without difficulties (item 1): $\overline{X} = 5.37$ and $s = 0.9$. These data indicate that students perceive that they can perfectly follow the contents of the subject through online classes.

Nevertheless, not all students thought that both teaching modalities (online and face-to-face) may be considered comparable as is shown by item 2. Here, respondents assessed

if they could follow the lesson in the same way online as they would do in the classroom. Results show a lower mean ($\overline{X}$ = 4.79) and a higher dispersion of responses (s = 1.35).

In item 3, with a mean of $\overline{X}$ = 5.03 and a standard deviation of s = 1.03, students stated that they can do all the proposed activities as if they were in the classroom.

Students indicated that the methodological strategy of the professors allowed them, in the online classes, to have a sense of belonging to their class even if they were at home and their classmates at university. This is shown by the results in item 4, where $\overline{X}$ = 5.04 and s = 1.14.

The results in item 5 underline that students felt more confident to participate when they attended online classes, which is reflected in a mean of $\overline{X}$ = 4.61 and a standard deviation of s = 1.27.

Concerning the pace of the lesson, results in item 6 ($\overline{X}$ = 2.82; s = 1.69) show that the methodological strategy used by the professors, namely repetitions and continuous encouragement, to participate for online students were not a great impediment for the lesson's development when attending in the actual classroom.

Item 7 explores if online classes could be an adequate solution not only in exceptional circumstances, such as those we are living in now. Data show a mean of $\overline{X}$ = 3.84 and a higher dispersion with s = 1.9.

Last, students were asked about the technical or other problems they experienced to adequately follow the online classes (item 8). Results show $\overline{X}$ = 2.44 and s = 1.72.

If we observe these data by specialization in the 2020–21 academic year (Table 3), results show, in general, similar values than those item by item. The Technology (TC) specialization shows the lowest means in almost all the items. In item 4, this difference is significant, meaning that TC students did not develop such a strong feeling of group belonging as the other students ($F_{3,91}$ = 5.427, *p* = 0.002, with $\overline{X}_{TC}$ = 4,20 and s = 1.36, opposite to $\overline{X}_{Total}$ = 5.04 and s = 1.14).In relation to the differences shown by group of class (Table 4), there are statistical differences in two items. Students in classes A and C reflected that they could follow online classes without any problems better than students in class B, whose mean in item 1 is significantly lower although all means show a good opinion about this issue ($F_{2,92}$ = 3.12, *p* = 0.049, with $\overline{X}_B$ = 5.10 and s = 1.19, opposite to $\overline{X}_{Total}$ = 5.37 and s = 0.91). On the other hand, students in class A considered that the lesson's pace was more affected by interference of the participation of their online classmates.

*4.2. Results for 2021–22 Academic Year*

The results for the 2021–22 academic year show that students value online classes favorably. The means are higher than 3.5/6 in all the items except in item 8. Actually, the scores are still relatively high, which shows that online classes are a good option and, in general, allow an adequate teaching–learning process. Specifically, they allow students to follow the lesson adequately, and they facilitate the sense of belonging to the group without major technical problems.

Concerning item 6 that addressed the issue of hindering the class' pace because of the hybrid methodology (students at home and students in the classroom), results show that students in the 2021–22 year did perceive this negative impact ($\overline{X}$ = 3.74 and s = 1.51).

Observing the results by the specialization in which the students were enrolled (Table 3), they show, in general, similar values in the different items. Technology students show lower means, but they are only significant for item 5. In this item, concerning the confidence to participate in class, Technology and Mathematics students have lower means ($F_{3,71}$ = 3.117, *p* = 0.031, with $\overline{X}_{TC}$ = 3.10 and s = 1.25; and $\overline{X}_{MAT}$ = 3.06 and s = 1.56, opposite to $\overline{X}_{Total}$ = 3.57 y s = 1.47).

If we attend to the analysis between the three classes (A, B, and C), no significant differences are observed for either of the items (see Table 4). The students' perception is statistically similar in all three groups.

**Table 4.** Results of the survey by class group. Mean (standard deviation) and Snedecor's F (ANOVA).

| Item | Year | Class A | Class B | Class C | Total | F<br>p |
|------|------|---------|---------|---------|-------|--------|
| Item 1. Online classes allowed me to follow the session without difficulties | 20–21 | 5.50 (0.61)<br>N = 36 | 5.10 (1.19)<br>N = 39 | 5.65 (0.59)<br>N = 20 | 5.37 (0.91)<br>N = 95 | 3.12<br>0.049 * |
| | 21–22 | 4.69 (1.16)<br>N = 26 | 4.38 (1.34)<br>N = 24 | 4.26 (1.51)<br>N = 27 | 4.44 (1.34)<br>N = 77 | 0.727<br>0.487 |
| Item 2. Online classes allowed me to follow the class the same way as face-to-face classes. | 20–21 | 4.97 (1.16)<br>N = 36 | 4.64 (1.51)<br>N = 39 | 4.75 (1.37)<br>N = 20 | 4.79 (1.35)<br>N = 95 | 0.567<br>0.569 |
| | 21–22 | | | Not applied | | |
| Item 3. In online classes, I was able to do all the proposed activities during class. | 20–21 | 5.03 (0.91)<br>N = 36 | 5.08 (1.1)<br>N = 39 | 4.95 (1.10)<br>N = 20 | 5.03 (1.03)<br>N = 95 | 0.100<br>0.905 |
| | 21–22 | | | Not applied | | |
| Item 4. In online classes, I was able to feel a sense of belonging to the group. | 20–21 | 5.03 (1.03)<br>N = 36 | 5.00 (1.30)<br>N = 39 | 5.15 (1.04)<br>N = 20 | 5.04 (1.14)<br>N = 95 | 0.117<br>0.890 |
| | 21–22 | 4.35 (1.29)<br>N = 26 | 4.04 (1.19)<br>N = 23 | 4.00 (1.52)<br>N = 27 | 4.13 (1.34)<br>N = 76 | 0.506<br>0.605 |
| Item 5. In online classes, I felt more confident to participate in class. | 20–21 | 4.67 (1.17)<br>N = 36 | 4.51 (1.43)<br>N = 39 | 4.70 (1.13)<br>N = 20 | 4.61 (1.27)<br>N = 95 | 0.198<br>0.821 |
| | 21–22 | 3.58 (1.42)<br>N = 26 | 3.64 (1.43)<br>N = 22 | 3.52 (1.60)<br>N = 27 | 3.57 (1.47)<br>N = 75 | 0.38<br>0.963 |
| Item 6. Following face-to-face classes with the participation of classmates from home hinders the class' pace. | 20–21 | 2.08 (1.25)<br>N = 36 | 3.46 (1.80)<br>N = 39 | 2.90 (1.71)<br>N = 20 | 2.82 (1.69)<br>N = 95 | 7.014<br>0.001 *** |
| | 21–22 | 3.58 (1.47)<br>N = 26 | 3.96 (1.36)<br>N = 23 | 3.70 (1.68)<br>N = 27 | 3.74 (1.51)<br>N = 76 | 0.390<br>0.679 |
| Item 7. Online classes are only a solution for the specific situation we are living in now. | 20–21 | 3.67 (1.84)<br>N = 36 | 4.13 (1.96)<br>N = 39 | 3.60 (1.93)<br>N = 20 | 3.84 (1.9)<br>N = 95 | 0.751<br>0.475 |
| | 21–22 | 4.46 (1.77)<br>N = 26 | 4.35 (1.40)<br>N = 23 | 4.30 (1.61)<br>N = 27 | 4.37 (1.59)<br>N = 76 | 0.072<br>0.930 |
| Item 8. During online classes, I had technical or other problems that hindered me following the class. | 20–21 | 2.14 (1.50)<br>N = 36 | 2.90 (1.86)<br>N = 39 | 2.10 (1.71)<br>N = 20 | 2.44 (1.72)<br>N = 95 | 2.378<br>0.098 |
| | 21–22 | 2.65 (1.60)<br>N = 26 | 2.78 (1.73)<br>N = 23 | 2.67 (1.82)<br>N = 27 | 2.70 (1.70)<br>N = 76 | 0.041<br>0.960 |

\* $p \leq 0.05$; \*\*\* $p \leq 0.001$.

### 4.3. Comparison of the Two Academic Years

When comparing the results obtained in the two analyzed years (Table 3), significant differences are observed in four of the six applied items. Students in the 2020–21 academic year, who received classes during the outburst of the pandemic, valued items 1, 4, and 5 higher than students in the 2021–22 year. They show a more positive opinion towards online teaching. On the other hand, students in the 2021–22 academic year perceived a more negative impact of online teaching on the pace of the class as can be observed in the results obtained in item 6.

### 5. Discussion

Due to COVID-19 pandemic, digital transformation has been accelerated in education, like in other fields. Although many universities had LMS and other technologies [45], there is no doubt that the pandemic has been a turning point in its adoption by teachers. Two years after the first global lockdown, it was demonstrated that the key to success in such transformation is not only the frequent use of these technologies. A high digital teaching competence is needed to incorporate hypermedia resources that promote the interactivity. Only with this will it be possible to motivate and promote students' interactions [2,15].

Even if the majority of higher education professors, as reported by [2,7], prefer face-to-face classes, they value the technology's potential and think it will play an important role in their future teaching career. Hence, as [14] elucidated, the pedagogical design in virtual teaching environments is crucial. It is, in fact, the key to success in students' learning processes. However, migrating face-to-face activities to online methods is not enough,

as teaching should also be reviewed and redesigned. Activities and planning must be modified to achieve the development of the implicated skills [15,16].

The professors involved in this study, who could be labeled as Leaders or Pioneers [14], had a high DTC according to the classifications presented in the reviewed studies [14,23,25]. A high DTC not only implies skillful handling of technologies but also requires having a criterion to select the most appropriate teaching technologies in each situation, allowing teachers to enrich their teaching, develop the digital competence of their students, and continue developing professionally [24,28].

The recommendations given by [16,18] for online classes were also shown to be useful for the hybrid model. Both professors followed them for preparing the classes and during and after the sessions. Online students participated in the activities, and there were tasks specifically designed to be presented online. The classes included online polls, quizzes, and surveys in which both classroom students and online students participated simultaneously.

Regarding our first research question, this work provided evidence in favor of the hybrid teaching model when DTC is high. While the study by [2] found some difficulties that limit a satisfactory learning experience (69%) (such as limited faculty digital competence (63%) and lack of adjustment of the online teaching style with the future courses they will take (62%)), the students in this study perceived that hybrid teaching facilitates learning.

Students, even when most of them are considered digital natives, do not always have the proper digital competence, hindering them sometimes in following the online sessions. This is not the case in this study, in which all students were able to follow the sessions remotely without difficulty. In fact, our results coincide with those of the study by [46] since no significant differences were found regarding the possibility to adequately follow the contents in an online class setting.

Furthermore, refs. [11,46,47] pointed out that one of the shortcomings of distance learning is the low participation of students from their connection devices. In contrast, with the sample considered for this work, made up of future secondary education teachers, results showed their capacity for maintaining focus and participating in the sessions. However, it is true that the sense of belonging to the group was lower for online teaching, which is in concordance with the results of [35,45].

Additionally, regarding the second research question about the different perception of the hybrid model depending on the academic year, data collected showed a better perception for online teaching during the first year (20–21) immediately after the COVID-19 crisis. This result is in accordance with [31], who explained that students perceived online teaching as a solution for the medical crisis. The decrease in the perception can be due to a lower sense of risk in the second year, highlighting the inconveniences of splitting the class in two and combining online and face-to-face class. At the same time, differences were appreciated both when considering the specialization and the different class groups. Specifically, Technology students showed the lowest values, having a significantly lower sense of belonging to the group. Both results highlighted the students' diversity in competences and preferences in learning styles. This draws attention to the need of addressing psycho-pedagogical aspects to facilitate their participation.

## 6. Conclusions

Most of the reviewed studies present a negative perception of the transition to hybrid or online models during the pandemic due to methodological difficulties, less student-teacher interaction, lack of teacher training in ICTs, or difficulties and lack of technical means [35–37]. All added to the difficulties generated by the stress and disorder in a time of pandemic [41]. However, data analysis indicates that, in this study, with this sample and under the circumstances of this experience, when teachers have both a high digital teaching competence [23] and means to apply it, the hybrid teaching model emerges as a viable solution. If teachers abide by the recommendations to maintain the effectiveness of synchronous classes [16,18], students are able to follow the online lessons, carrying out the same activities virtually as on site. Furthermore, they develop a feeling of belonging

to their class and are more confident to participate through the provided channels. The lessons' pace is not affected when having to attend to remote and on-site students.

In addition, the students that participated in this experience, who are future teachers, believe that this methodology could be useful even when this exceptional situation comes to an end. Furthermore, students closed the digital gap in app and equipment usage and in development of telematic networks.

We are aware that this study does not set an ending at this point, and therefore, our next step should be oriented to optimize the model. Applying it to other students from the same master course and from different ones during the following academic years when the pandemic situation has ended will allow us to widen our knowledge about students' perceptions, the relevance of faculty's DTC, and hence develop a better adapted hybrid model.

Technologies that allow distance learning have been implemented in our university thanks to the COVID-19 crisis during the last two years, and we believe in the necessity to continue exploring their possibilities. However, it is necessary to continue training professors so that they are more confident in the use of ICTs and so this becomes a support for methodologies and not a barrier. Educational technologies are constantly changing, and professors must develop their DTC to be prepared for future challenges.

**Author Contributions:** J.L.M.-N., conceptualization, formal analysis, and writing—original draft preparation; J.L.B.-R., conceptualization, methodology, formal analysis, and writing—original draft preparation; S.S.-M., conceptualization, methodology, formal analysis, data curation, and writing—original draft preparation; I.P.-L., writing—review and editing; visualization, and supervision; A.C.R., visualization and supervision; C.N.-d.-R., conceptualization, methodology, formal analysis, data curation, and writing—original draft preparation. All authors have read and agreed to the published version of the manuscript.

**Funding:** This research received no external funding.

**Institutional Review Board Statement:** Not applicable.

**Informed Consent Statement:** Not applicable.

**Data Availability Statement:** Not applicable.

**Acknowledgments:** PROFICIENCYIn+Edu: Formación colaborativa en competencias docentes para la inclusión y la excelencia. MINECO/FEDER RTI2018-096761-B-I00.

**Conflicts of Interest:** The authors declare no conflict of interest.

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
