# Peer review of "Teaching in Secondary Education Teacher Training with a Hybrid Model: Students’ Perceptions"

_sustainability, doi:10.3390/su14063272_

Round 1
Reviewer 1 Report
This article is devoted to the study of the application of the hybrid model in the process of processing. The topic of the article is relevant and important in modern conditions of education. Methods correspond to the goal. The conclusions reflect the content of the article. However, in my opinion, the conclusions can be expanded and more detailed. In this case, the results will be more clearly shown, and not hidden in the text of the article. Moreover. one of the keywords in this article is "Teacher digital competence". And this aspect should be more clearly presented in the article and the conclusions of the article.
Reviewer 2 Report
Although the idea underlying this piece of research is interesting, I believe the paper cannot be published as is due to a serious of factors:
- The literature review lacks specificity. It is too general, not specifically related with the research aims: the evaluation of students’ perception in a ‘hybrid’ mode combining online and FTC learning in a teacher training program. There is today a good amount of works published on pre-service teachers perception of FTF/online/blended/bimodal learning from different perspectives: motivation, distraction, interaction, specific tools, etc. In their introduction, the authors adopt a very general view about the alleged requirements for effective online teaching and the use of technology in general but they do not focus on works directly related with this ‘hybrid’ model. So, it might be more interesting to concentrate on some specific issues related with previous literature about pre-service teachers’ attitudes and perception in order to explain the research need. It is not clear this paper aims to bridge a research gap in this sense.
- There is a need to clarify and better explain some key concepts and principles used throughout the paper, particularly what is the difference between a ‘hybrid’ model and ‘bi-modal’, if any? What is the difference between blended and ‘hybrid’ or ‘bi-modal’, which seem to be sometimes used interchangeably? What is ‘ryhtm’ in ‘maintaining subject’s (sic) rhythm’? (246).
- There is no information about the context and the results are clearly limited to such context, for example when the authors state that ‘This would also be the moment to ask them to turn on their cameras and try to keep them on, as long as the technical conditions permit it’ (107-108), they should consider this could depend not only on ‘technical conditions’ but also on other factors such as the existence of specific regulation (data protection, use of personal cams), etc. More details about the research context and its implications are needed.
- There is a good amount of claims which are not fully proven. In fact, the authors either seem to assume uncritically some statements from other authors which could be contradicting considering current literature (see a below) or seem to adopt an overenthusiastic view about their own research (b). For example:
- Among the conditions to be fulfilled according to another author they indicate ‘Designing assessment tasks and strategies with humane evaluation and marks, trying to keep technological mediation to a minimum’. This should be considered from a more critical perspective such as ‘According to this author …’
- ‘the RTSC subject was assessed by the students and its professors achieved approval values greater than 90% in the three classes (A, B and C) where the subject was taught. This is an exceptionally high evaluation’ (230-232). What is ‘Exceptionally high’ here? Compared to what? Any previous literature?
- The learning objectives are neither defined nor clearly correlated with the research findings (discussion, conclusion).
- The research instruments, particularly the ad-hoc survey, are weak. The authors state that ‘This survey consists of 36 questions with a 6-level Likert scale from completely agree (6) to completely disagree (1)’ (235-236). Then, they express ‘These items, designed ad hoc, refer to different aspects about online teaching, compared to face-to-face teaching’ (239) and include a list of what they also call ‘aspects’. There are several issues here: Why did they create their ad-hoc survey instead of using a validated one about students’ perception in previous works (no reference to previous works on this issue have been made in the introduction)? What is the reliability of this ad-hoc survey (no Cronbach’s alpha is provided)? Where are the 36 questions which make up part of this ad-hoc survey? This is critical as they have not used a previously validated scale so we have no data about its validity. Why do they say ‘36 questions’ if they mention ‘six aspects’ which are also called ‘items’ (239-248)? These six items or aspects are not ‘questions’, so could it be that they ad-hoc survey was made up of 36 items in the form of ‘questions’ organized into 6 dimensions they call ‘aspects’ (or ‘items’)? More importantly, they mention this ad-hoc survey consisted of ‘36 questions’ based on a ‘6-level Likert scale’, so were these items questions or actually statements (‘agree-disagree’)? We do not have any information on this, there is no validated scale used in previous research and no 36-items (questions) in the appendix to evaluate this instrument (and measures taken to avoid survey bias, particularly in statements based on students’ self-perception), so it seems difficult to agree with the findings.
- The six aspects or items or questions included in this ad-hoc survey need to be further clarified and clearly defined (page 6). What do ‘difficulties’ mean in aspect 1 and 8? What does ‘Following online classes, the same way as face-to-face classes’ (2) mean? What does ‘sense of belonging to a group’ mean? How was ‘participating in class’ rated or was it self-perceived by students? And what does ‘rhythm’ mean in ‘Maintaining the subject’s rhythm’? They could have defined and explained all these concepts (6 aspects or items) in the introduction in reference to previous works.
- The conclusions include several statements which are not proven. For example:
‘results show their capacity for maintaining focus and participating in the sessions.’ (413) How do they know if this seems to be self-perceived?
‘The lessons rhythm is not affected when having to attend remote and on-site students’ (429) How can this be proven?
‘Also, students have closed the digital gap’ (432) How can this be proven? There is no reference to previous data on technology affinity before the experiment among the same subjects or to previous research in the field
- The images included in Figure 1 are not relevant.
In short, there seems to be a lack of specificity about the literature review, the research objectives and the actual research carried out. Also, the design of the instrument (ad-hoc survey) used for the collection of data seems to greatly undermine the validity of the conclusions
Reviewer 3 Report
Dear authors, cordial greetings, thank you for allowing me to review your manuscript.
Some recommendations as an opportunity for improvement so that your manuscript can be published.
In the instruction section there are many paragraphs with statements without theoretical support, please support it with an author.
The manuscript does not establish in its introduction section, the research gap and much less the research question.
There are very long paragraphs, which tends to be confusing, it is recommended to shorten the paragraphs so that the reader has a better appreciation of the manuscript.
How this work aims at sustainability does not appear in this section either, without forgetting the special and criteria of the journal.
The manuscript lacks a literature review section, so it is recommended that you give an answer to your keywords and that this is visualized in a section of the future article, I recommend that you can review the following published works:
Jiménez-Bucarey, C.; Acevedo-Duque, Á.; Müller-Pérez, S.; Aguilar-Gallardo, L.; Mora-Moscoso, M.; Vargas, E.C. Student's Satisfaction of the Quality of Online Learning in Higher Education: An Empirical Study. Sustainability 2021, 13, 11960. https://doi.org/10.3390/su132111960. https://doi.org/10.3390/su132111960
Rojo-Ramos, J.; Manzano-Redondo, F.; Adsuar, J.C.; Acevedo-Duque, Á.; Gomez-Paniagua, S.; Barrios-Fernandez, S. Spanish Physical Education Teachers' Perceptions about Their Preparation for Inclusive Education. Children 2022, 9, 108. https://doi.org/10.3390/children9010108. https://doi.org/10.3390/children9010108
Rojo-Ramos, J.; Manzano-Redondo, F.; Barrios-Fernandez, S.; Garcia-Gordillo, M.A.; Adsuar, J.C. A Descriptive Study of Specialist and Non-Specialist Teachers' Preparation towards Educational Inclusion. Int. J. Environ. Res. Public Health 2021, 18, 7428. https://doi.org/10.3390/ijerph18147428
UNESCO. Incheon Declaration. Towards Inclusive and Equitable Quality Education and Lifelong Learning for All; UNESCO: Paris, France, 2015. https://revistaeducacioninclusiva.es/index.php/REI/article/view/48
In the section on methodology, it is one of the most interesting of this research project, however it would be interesting to place if the instrument used was validated by an expert, or if it was used by an existing one, which is recommended to place the reference.
As for the results, they are very well done, which shows an interesting content.
In the discussion section, it is recommended to include the findings of this research project and to discuss them with works carried out by other authors showing the significant change existing in the spaces of university knowledge.
It would be interesting to expand the conclusion section by showing in more detail the prospective of the research project.
The references are straight in information, some of them must respect the editorial criteria and their forms.
Many successes.
Round 2
Reviewer 2 Report
Thanks for your revised version.
Reviewer 3 Report
Dear authors greetings.
I appreciate the opportunity to review your work. Best regards.